# Disruption of HaVipR1 confers Vip3Aa resistance in the moth crop pest *Helicoverpa armigera*

Andreas Bachler[1]*, Amanda Padovan[1], Craig J. Anderson[2], Yiyun Wei[3], Yidong Wu[3], Stephen Pearce[1], Sharon Downes[4], Bill James[1], Ashley E. Tessnow[5], Gregory A. Sword[5], Michelle Williams[1], Wee Tek Tay[1], Karl H. J. Gordon[1], Tom K. Walsh[1]

1 CSIRO, Black Mountain Laboratories, Acton, Australian Capital Territory, Australia, 2 MRC Human Genetics Unit, Institute of Genetics and Cancer, University of Edinburgh, Edinburgh, United Kingdom, 3 College of Plant Protection, Nanjing Agricultural University, Nanjing, China, 4 CSIRO, Myall Vale Laboratories, Kamilaroi Highway, Narrabri, New South Wales, Australia, 5 Department of Entomology, Texas A&M University, College Station, Texas, United States of America

* Andy.Bachler@csiro.au

**Academic editor:** Louis Lambrechts, Institut Pasteur, FRANCE

## Abstract

The global reliance on *Bacillus thuringiensis* (Bt) proteins for controlling lepidopteran pests in cotton, corn, and soybean crops underscores the critical need to understand resistance mechanisms. Vip3Aa, one of the most widely deployed and currently effective Bt proteins in genetically modified crops, plays a pivotal role in pest management. This study investigates the molecular basis of Vip3Aa resistance in Australian *Helicoverpa armigera* through genetic crosses, and integrated genomic and transcriptomic analyses. We identified a previously uncharacterized gene, LOC110373801 (designated *HaVipR1*), as potentially important in Vip3Aa resistance in two field-derived resistant lines. Functional validation using CRISPR/Cas9 knock-out in susceptible lines confirmed the gene's role in conferring high-level resistance to Vip3Aa. Despite extensive laboratory selection of Vip3Aa-resistant colonies in Lepidoptera, the biochemical mechanisms underlying resistance have remained elusive. Our research identifies HaVipR1 as a potential contributor to resistance, adding to our understanding of how insects may develop resistance to this important Bt protein. The identification of *HaVipR1* contributes to our understanding of potential resistance mechanisms and may inform future resistance management strategies. Future work should explore the biochemical pathways influenced by *HaVipR1* and assess its interactions with other resistance mechanisms. The approach utilized here underscores the value of field-derived resistant lines for understanding resistance in agricultural pests and highlights the need for targeted approaches to manage resistance sustainably.

**Data availability statement:** All raw sequence data generated in this project are available from BioProject PRJNA1119665.

**Funding:** We gratefully acknowledge the support received from various organizations and funding bodies. AET was supported by a travel grant (CLW1602) from the Cotton Research & Development Corporation (CRDC) and (16-413) from Cotton Incorporated. CJA was funded by the Commonwealth Scientific and Industrial Research Organisation (CSIRO) for his Postdoctoral research. This work was supported by several projects from the CRDC, including CSE1201 (original), CSE112, CSE104C, CSE0002, and CSE1103. The funders had no role in study design, data collection and analysis, decision to publish, or preparation of the manuscript.

**Competing interests:** The authors have declared that no competing interests exist.

**Abbreviations :** Bt, *Bacillus thuringiensis*; CAAS, Chinese Academy of Agricultural Sciences; DGE, differential gene expression; PM, peritrophic matrix; RAD-seq, Restriction site-associated DNA sequencing; VIPs, Vegetatitve Insecticidal Proteins.

## Introduction

Proteins derived from *Bacillus thuringiensis* (Bt), primarily Cry toxins, have been used in crops to control lepidopteran pests since 1996 [1]. The use of single toxins for control of pests rapidly selects for resistance and so development of alternative types of toxins is required. One of the alternative candidates developed are the Vegetatitve Insecticidal Proteins (VIPs), with a specific member (Vip3Aa) of these proteins showing high effectiveness against a broad range of lepidopteran pests [2]. While both Cry and VIP toxins cause disruption in the gut upon ingestion by a target pest current evidence suggests distnict modes of action, with resistance to one toxin typically not conferring resistance to the other type of toxin [3]. This provides an option for pest management in cases where resistance to Cry toxins is increasing, and subsequently in the United States of America (USA) and South America, large areas of corn and cotton with Vip3Aa have been planted since 2008 [4]. In Australia, Vip3A was deployed in 2016 in combination with Cry1Ac and Cry2Ab in cotton (Bolgard III), primarily to control the global pest Cotton Bollworm, *Helicoverpa armigera*, and an Australian local species *Helicoverpa punctigera* [1]. A primary concern for the continued effectiveness of Vip3Aa in high-impact crops globally is the rapid identification and management of resistance to Vip3Aa.

Field-derived resistance to Vip3Aa has been detected in a number of lepidopteran species, although, to date, there have been no verified field failures directly linked to Vip3Aa resistance [1,5,6]. Early studies on resistance alleles in field populations and the subsequent establishment of laboratory-resistant lines have provided valuable insights into the genetic and phenotypic characteristics of Vip3Aa resistance [7,8]. Prior to the commercialization of Vip3Aa, $F_2$ screens detected low levels of resistance alleles in Australian *H. armigera* (2.7% frequency) and *H. punctigera* (0.8%), and these resistance frequencies have remained relatively stable from 2009 to 2023, despite widespread deployment of the toxin in cotton crops [1,8]. In North America, $F_2$ screens of *Helicoverpa zea* from Texas estimated the frequency of resistance alleles at 0.65% [5], while an independent study from Louisiana found a slightly higher frequency of 1.49% [9]. Similarly, *S. frugiperda* populations in Brazil, Louisiana, and Florida also exhibited low frequencies of resistance allele frequencies [10,11]. These early data suggest that resistance to Vip3Aa is relatively rare in the field, but continuous monitoring is critical, especially as new reports of increasing resistance emerge, such as for *H. zea* in the USA [5]. Molecular monitoring can also be useful in identifying potentially novel modes of resistance, such as the potential for multigenic tolerance to Vip3A in *H. zea* [12], which may pose a growing challenge for pest management.

Despite the widespread use of Vip3Aa and the identification of numerous resistant phenotypes from field populations, the specific genes and molecular pathways involved in conferring resistance to this toxin remain largely unknown. In an effort to better understand the genetic basis of resistance, insect lines resistant to Vip3Aa have been isolated from both field populations and laboratory-selected strains. Selection for Vip3Aa resistance in *Heliothis virescens* over 13 generations

resulted in a 2,040-fold resistance ratio relative to susceptible individuals [13]. Similar laboratory selection experiments have demonstrated high resistance ratios in other species, such as *S. frugiperda*, which exhibited resistance ratios exceeding 9,800-fold [6] and 3,200-fold [14], as well as in *H. zea*, which showed >588-fold resistance [15]. Most cases of Vip3Aa resistance in Lepidoptera have been shown to be recessive, likely mediated by a single autosomal gene, and without cross-resistance to Cry1Ac or Cry2Ab [7,8,16]. As a model for high-throughput screening, *Sf9* cell lines, derived from *S. frugiperda* ovarian tissue, have shown sensitivity to Vip3Aa, and their interaction with Vip3Aa has been studied for potential insights into the mode of action and resistance mechanisms [17,18]. Interestingly, the binding sites for Vip3A toxins are generally distinct from those of Cry1 and Cry2 toxins, with the exception of Cry1Ia in *Spodoptera eridania* [19], which may explain the absence of cross-resistance between Vip3Aa and Cry1- or Cry2-resistant lepidopteran colonies [20]. Further investigations into dual-resistant individuals have shown that resistance to Vip3Aa and Cry toxins are not linked in *H. armigera* and *H. punctigera* [21], suggesting these toxins may operate through distinct modes of action. However, while resistance mechanisms for Vip3Aa in most species remain unclear, recent studies have identified potential candidates involved in resistance. For example, a transcription factor, *SfMyb*, was implicated in mediating resistance in a lab-selected Vip3Aa strain, although the specific targets of *SfMyb* that lead to resistance remain unresolved [22]. Furthermore, the first report of a chitin synthase gene being involved in Vip3Aa resistance in *S. frugiperda* was recently made [23], providing a potential avenue for future research on the genes or pathways involved in Vip3Aa resistance.

The identification of a gene associated with Vip3Aa resistance in *H. armigera* contributes to our understanding of potential resistance mechanisms to this Bt toxin. By leveraging advanced genomic tools, including transcriptome sequencing, CRISPR/Cas9 gene editing, and long-read sequencing, this study provides insights into potential genetic factors associated with resistance and also highlights the diverse mechanisms—such as transposable element insertions—that can disrupt resistance-associated genes. These findings may contribute to future monitoring approaches and resistance management strategies for *H. armigera.*

*Summary:* The first [24] and subsequent [25] *H. armigera* genomes have opened new avenues of research into this costly pest, including the modes of action and mechanisms of resistance to Bt proteins. In this work, we describe the characterization of a Vip3A-resistant lines ("Ha85", previously identified as SP85 by [26]), conducted genetic mapping of Vip3A resistance, and identified a single candidate locus associated with resistance. Using transcriptome sequencing, we identified the gene responsible for Vip3A resistance. We confirmed the role of this gene by employing CRISPR/Cas9 to knock it out, successfully generating a highly resistant Vip3Aa phenotype. Using long-read sequence, we also identified the insertion of a transposable element in the first intron of the candidate resistance gene in another Vip3A-resistant line (Ha477), which is allelic for resistance with Ha85, indicating multiple ways this gene can be disrupted in Vip3Aa field-derived resistant lines.

## Results

### Genomic mapping of resistance loci

To identify genomic regions under selection for Vip3A resistance, progeny from a female-informative backcross were analyzed using Restriction site Associated DNA sequencing (RAD-seq). Progeny were either exposed to a Vip3A surface overlay (selected) or left untreated (control). A total of 32,920 loci were identified in the treated ($n = 20$) and control ($n = 20$) populations using STACKS analysis with the HaSCD2 reference. Smoothed $F_{st}$ and Phi ($\Phi_{st}$) were assessed for all loci. Invalid values (such as those with negative $F_{st}$) were set to 0. All chromosomes had between 400 and 2,000 valid loci able to be assessed (see Table A in S1 Text for raw counts). The mean value for $F_{st}$ and $\Phi_{st}$ was assessed in 1 megabase pair (mbp) genomic windows for each chromosome. The results indicated a single chromosome (chromosome 2) was under selection for Vip3A resistance in the selected population (Fig 1, S1 Data).

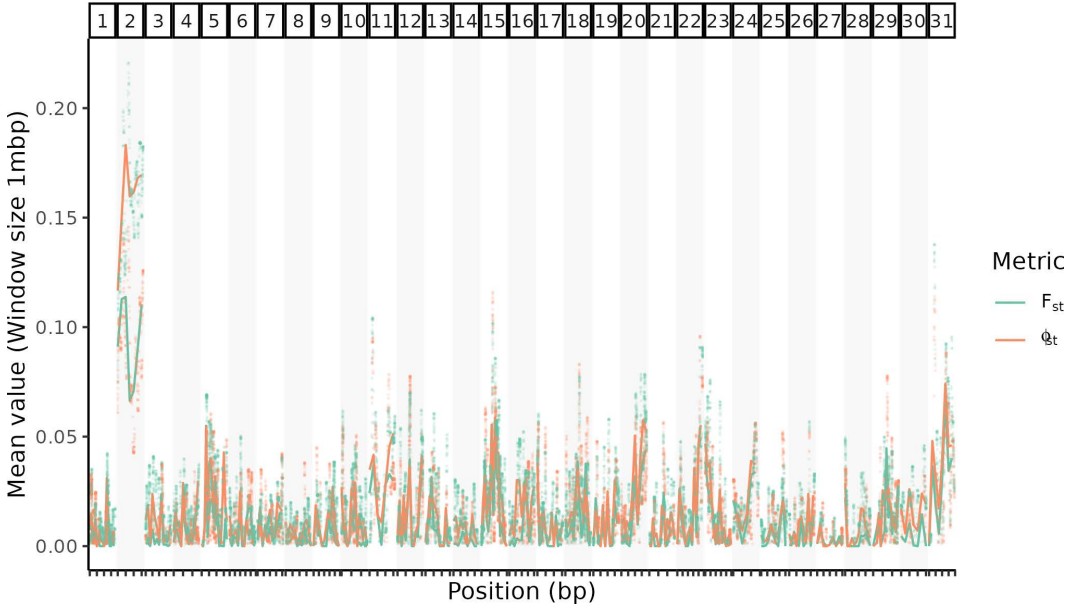

**Fig 1. Mapping of resistance-associated loci from a female-informative cross indicates a single chromosome under selection for Vip3A resistance.** $F_{st}$ and Phi ($\Phi_{st}$) values were derived from RAD-Seq data to identify genomic regions under selection for Vip3A resistance in progeny from a female-informative backcross. Genetic markers across all chromosomes were assessed using STACKS, with raw $F_{st}$ and $\Phi_{st}$ values plotted alongside mean values calculated in 1mbp genomic windows per chromosome. Progeny were either exposed to a Vip3A surface overlay (selected) or left untreated (control). Data available in S1 Data.

### Transcriptomic analysis of Vip3A-resistant lines

Differential gene expression (DGE) between Vip3A-resistant (Ha85) and susceptible (GR) lines was assessed using pooled RNA from mid-gut tissues. Analysis was performed with DeSeq2, and genes with a total combined count of less than 100 reads were omitted, resulting in the assessment of 6,790 genes (mapping statistics for all samples present in Table B in S1 Text). The results are visualized in a Manhattan plot, highlighting some differential expression across the 31 *H. armigera* chromosomes, with the most significant gene being LOC110373801 on chromosome 2, annotated as "Uncharacterized protein." (Fig 2A and S2 Data). We refer to this as *HaVipR1* throughout this manuscript. The expression of this gene was almost non-existent in the resistant Ha85 line versus the susceptible strain (Fig 2B and S2 Data ) (raw counts and summary statistics generated from DeSeq2 for the HaVipR1 gene are available in Table C in S1 Text). The differential expression observed in RNA-Seq was validated using qPCR analysis on mid-gut samples from the Vip3A-resistant (Ha85), susceptible (GR), and an additional resistant line (Ha477, allelic for resistance to Ha85). The transcription levels of LOC110373801 as assessed using qPCR were significantly down-regulated in both resistant lines compared to the susceptible line, corroborating the DGE results (Fig 2C and S2 Data).

### Generation and validation of a *HaVipR1* knockout strain using CRISPR/Cas9

Fresh eggs were injected individually with one nanolitre mix of the Cas9 protein and the sgRNA of *HaVipR1*. Among the injected eggs, 14.6% (135/925) of them hatched. Among the 135 neonates, 57.8% (78/135) of them developed into adults ($G_0$). The $G_0$ moths were individually crossed with SCD moths (single-pair mating) to produce the next generation ($G_1$). Forty-eight single-pair families produced fertile progeny. Six to ten larvae were randomly selected from each single-pair family to check if there were inherited indel mutations, and 30 out of the 48 single-pair families had at least one indel mutation created by Cas9 using PCR amplification. When a cluster of double peaks was observed around

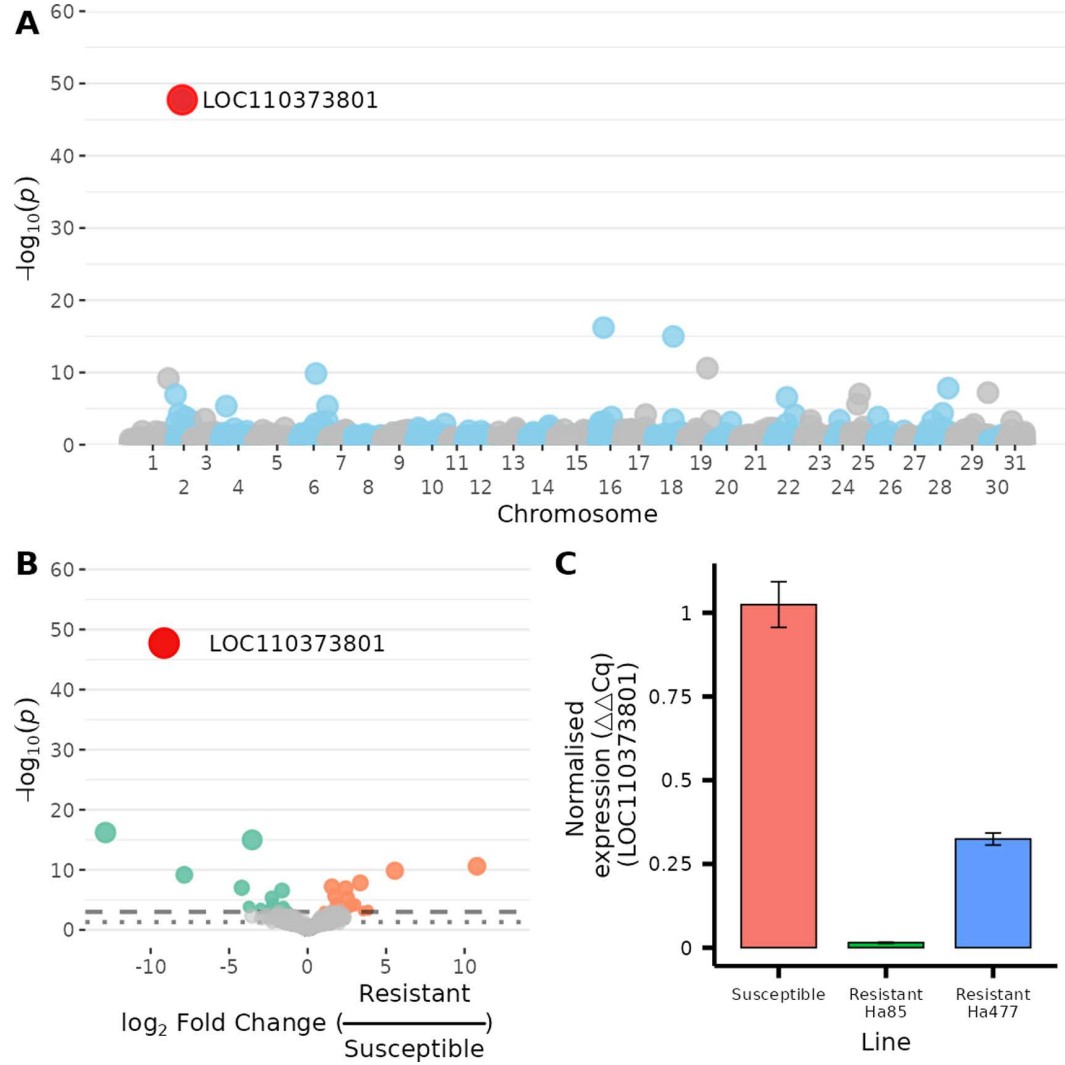

**Fig 2. Differential gene expression in Vip3A-resistant vs. susceptible *Helicoverpa armigera* lines.** (A) Pooled RNA mid-gut tissues from a resistant (Ha85) and susceptible (GR) line were analyzed for differential gene expression analysis with DeSeq2. A Manhattan plot of assessed genes was generated for all assessed genes across the 31 *H. armigera* chromosomes. The gene exhibiting the most significant differential expression (LOC110373801, annotated as "Uncharacteriszed protein") is highlighted. (B) Visualization of differential expression between the resistant and susceptible lines using a volcano plot. Significance thresholds are indicated by dashed (p < 0.01) and dotted (*p* < 0.05) lines. (C) Quantitative PCR (qPCR) analysis of the transcription level of LOC110373801 in mid-gut samples from three distinct groups of *H. armigera*: the same Vip3A-resistant (Ha85) and susceptible (GR) lines used in the transcriptome analysis, and an additional Vip3A-resistant line (Ha477), allelic for resistance to Ha85. Transcription levels were normalized to the reference gene EF-1α. Data available in S2 Data.

the Cas9-induced DSB cutting site (Fig A in S1 Text), PCR products were TA-cloned and sequenced by Life Technology (Shanghai, China) to determine the exact indel mutation type. Most abundant indel mutations flanking the target site were listed in Fig A in S1 Text for the $G_1$ larvae from the single-pair families ($G_0 \times$ SCD).

Fifty-three pupae from one of 30 single pair families were selected for genotyping at exon 3 of *HaVipR1* using exuviates of the final instar larvae (as in [27]). Among the 53 individuals sequenced, 20 were heterozygous mutants of 2-bp deletion, 14 were heterozygous mutants of 16-bp deletion, 7 were heterozygous mutants of 3-bp deletion, 1 was heterozygous

mutants of 1-bp deletion, and 11 were wild-type homozygotes. To transmit the mutant allele to $G_2$, we mass-crossed the 6 male moths and 8 female moths which were heterozygous for the 16-bp deletion in exon 3 of *HaVipR1*.

Neonates of $G_2$ were screened for resistance with a discriminating concentration of 50 µg Vip3Aa/cm$^2$, and 19.1% survived (78/408). Pupae of the 48 survivors and 52 untreated larvae were genotyped for the 16-bp deletion in exon 3 of *HaVipR1* using exuviates of the final instar larvae. For the untreated 52 larvae, the expected 1:2:1 ratio was observed: homozygous wild type (12), heterozygotes (28), and homozygous mutant type (12). In contrast, all of the 48 survivors were identified as homozygous for the 16-bp deletion exon 3 of *HaVip3R1*. To statistically confirm this association, we performed a chi-squared test of independence (degrees of freedom = 2). The observed and expected genotype frequencies are summarized in Table 1. The chi-squared test resulted in a $\chi^2$ value of 61.538 and a corresponding *p*-value of 4.336e−14, which is smaller than the threshold for significance of 0.05, rejecting the null hypothesis of independence. These results provide strong statistical evidence of a significant association between the Vip3Aa resistance phenotype and the 16-bp deletion mutation of *HaVipR1*. The 48 survivors were pooled and mass-crossed to generate a Vip3Aa-resistant strain, named 548KO2.

Surface-treatment bioassay measuring resistance to Vip3Aa was conducted for both the susceptible strain (SCD) and the knockout strain (548KO2). For the susceptible strain, the $LC_{50}$ was 0.32 (95% FL 0.21–0.55) µg/cm$^2$, Slope = 2.54 (SE: 0.26). For the knockout strain, mortality was 20.8 (5/24 tested) at the highest dose of surface treatment tested (500 µg/cm$^2$), and so $LC_{50}$ was unable to be determined. Both the Ha85 and Ha477 Vip3Aa resistant strains had been previously assessed for Vip3Aa resistance and displayed low mortality at the highest Vip3Aa concentration able to be tested (220 µg/cm$^2$) [8] The $LC_{50}$ values are similar between the SCD strain used to generate the knockout (conducted in China with the *H. armigera armigera* sub-species, $LC_{50}$ 0.32 µg/cm$^2$), and the susceptible line used in Australia (which is of the *H. armigera conferta* sub-species, $LC_{50}$ 0.55 µg/cm$^2$). Assuming that the $LC_{50}$ is the maximum concentration tested for the Ha85, Ha477, and the 548KO2 knockout strains (220 µg/cm$^2$), we can calculate the resistance ratio of the 528K02 knockout as 687-fold and for the Ha85 and Ha477 lines as 660-fold.

To characterize the inheritance pattern of resistance, we performed reciprocal crosses between the resistant 548KO2 strain and the susceptible SCD strain. Following established approaches for determining inheritance patterns in Vip3Aa resistance [16,22], we tested the parental and $F_1$ progeny using a concentration of 4 µg/cm$^2$, which falls below the standard discriminating dose of 10 µg/cm$^2$ used to distinguish resistant and susceptible *H. armigera*. At this concentration, the resistant 548KO2 strain showed 100% survival (96/96), while both the susceptible SCD strain and the reciprocal $F_1$ progeny showed complete mortality (0/96 for each cross). The dominance parameter (*h*) calculation yielded a value of 0, providing strong evidence for recessive inheritance. The complete mortality of heterozygotes at a concentration above the susceptible $LC_{50}$, yet well below the resistance threshold of the resistant strain, supports the conclusion that resistance is functionally recessive.

## Phylogenetic and functional analysis of *HaVipR1* gene

**Phylogenetic conservation.** The protein product of the *HaVipR1* gene (LOC110373801) was analyzed for phylogenetic conservation across diverse insect taxa. A phylogenetic tree was constructed using sequences identified via

**Table 1. Observed genotyping results following selection of CRISPR disrupted mutants (*r* indicates the 16-bp deletion, *S* indicates the susceptible wild-type).**

|  | *SS* | *Sr* | *rr* | *Total* |
|---|---|---|---|---|
| Survivors (48) | 0 | 0 | 48 | 48 |
| Control (52) | 12 | 28 | 12 | 52 |
| Total | 12 | 28 | 60 | 100 |

BlastP and analyzed with IQ-TREE (Fig 3). The generated phylogenetic tree revealed that the *HaVipR1* gene is relatively conserved within moth and butterfly groups, forming a distinct clade separate from other insects, such as beetles and mosquitoes, which showed higher divergence in sequence homology and functional annotations.

**Functional annotation.** Functional domains within the *HaVipR1* protein sequences were assessed using InterProScan, and annotations from the SUPERFAMILY and PANTHER databases were included for each sequence and indicated the presence of conserved thyroglobulin type-1 (SSF57610/IPR036857) and protease inhibition (PTHR12352/IPR051950) domains across almost all sequences. The mosquito clade included an additional domain relating to the growth factor receptor domain (SSF57184/IPR009030). PANTHER GO terms indicated involvement in the biological process of cell-matrix adhesion (GO:0007160), with cellular components including the basement membrane (GO:0005604) and extracellular space (GO:0005615), while no specific molecular functions were identified.

### Structural variation in resistant and susceptible lines

Genetic analysis of the *HaVipR1* gene in Vip3A-resistant and susceptible *H. armigera* lines revealed significant structural variation (Fig 4). In the resistant Ha85 line, a 149 bp deletion in the 5′ untranslated region (5′ UTR) was identified through visual inspection of the sequence alignment of genomic sequencing data from the Ha85 line. This deletion removes the original Kozak sequence present for this gene (ACA-ATC-AAA-**ATG**) and replaces it with a new sequence (ACG-TCA-GAA-**ATG**) (Bold indicates start codon of gene). In the new sequence, a key nucleotide at the −3 position from the start codon is altered, as well as substitutions for some of the remaining bases. Disruption of the Kozak sequence can lead to disregulation of gene function through changes in protein translation [28]. No other major effect variant was identifiable from inspection of the alignment. In the resistant Ha477 line, a large transposable element was found, extending intron 1 by ~10 kbp and impacting gene splicing, with spliced alignments identified near the 3′ end of the repeat-rich region to exon 2 of the *HaVipR1* gene (see Fig B in S1 Text).

### Discussion

Globally, the control of lepidopteran pests in cotton, corn, and soybean heavily relies on Bt proteins, with increasing prevalence of Vip Bt toxins [29]. In this study, we identified the *HaVipR1* gene (LOC110373801) as being associated with Vip3Aa resistance in two field-derived resistant lines of *H. armigera*. This identification was achieved through a combination of genetic crosses, genomic, and transcriptomic analyses. We then used CRISPR to knock this gene out in susceptible *H. armigera* lines conferring resistance and confirming a role for this gene in Vip3A resistance.

Our analyses suggest this gene differs from previously reported Vip3A-associated genes. In *S. frugiperda*, two candidate Vip3A resistance genes have recently been reported (the transcription factor "SfMyb" [22] and the chitin-synthase gene "SfCHS2" [23]). The domains identified in the *HaVipR1* protein show it does not have a similar function to either of these two proteins. Analysis of the homologs of the *S. frugiperda* Vip3A resistance genes in the susceptible and resistant *H. armigera* transcriptome data also shows no statistically significant association with Vip3A resistance (Fig C in S1 Text, Table D in S1 Text, S2 Data). Further assessment of *HaVipR1* in relation to other known Vip3A receptor genes, generally studied in Sf9 cell lines, revealed no significant association in the resistant lines (Fig C in S1 Text, Table D in S1 Text, S2 Data). These receptor genes include ribosome S2 protein 3, tenascin-like glycoprotein 4, fibroblast growth factor receptor-like protein 5, and scavenger receptor class C-like protein. The expression of the *HaVipR1* gene appears limited to the mid-gut and to be extracellularly located (assessed in both *H. armigera* (Fig D in S1 Text, S3 Data, originally generated by [Ioannidis et al]) and for the homolog of HaVipR1 in *Bombyx mori* (Fig E in S1 Text, S4 Data). The homolog of *HaVipR1 in S. frugiperda* does not appear to be expressed endogenously in Sf9 cell lines, which are ovarian derived (Fig F in S1 Text, S5 Data). While Sf9 cell lines do appear to be susceptible to Vip3Aa [18], the translation of findings from Vip3Aa studies from in vitro cell-line studies to whole animal models is sometimes not supported, as seen in the case of Sf-FGFR and Sf-SR-C which were found to be receptors in Sf9 cell lines but CRISPR-mediated disruption in whole

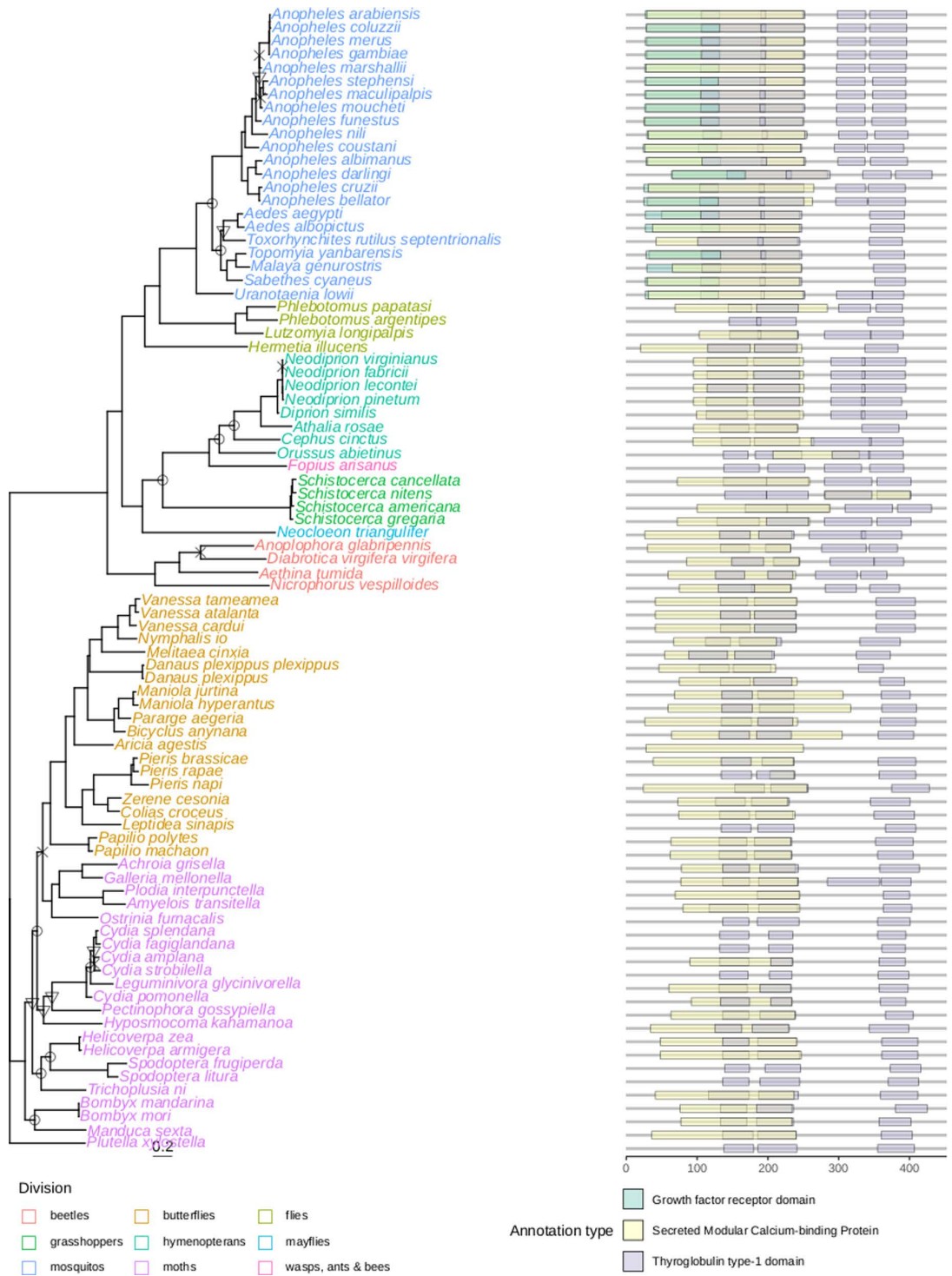

**Fig 3. Phylogenetic conservation and domain analysis of the HaVipR1/LOC110373801 gene across insect taxa.** (A) Phylogenetic analysis of HaVipR1 using BlastP to identify top protein hits, aligned with MUSCLE. The tree was constructed with IQ-TREE, using ModelFinder for model selection and 1,000 Ultrafast bootstraps for node support. Bootstrap support is noted as: < 70% (✕), 70%–80% (▽), and 80%–90% (○); nodes with >90% support are not shown. (B) Functional domain analysis via InterProScan, with annotations from SUPERFAMILY and PANTHER databases for each protein in the phylogeny.

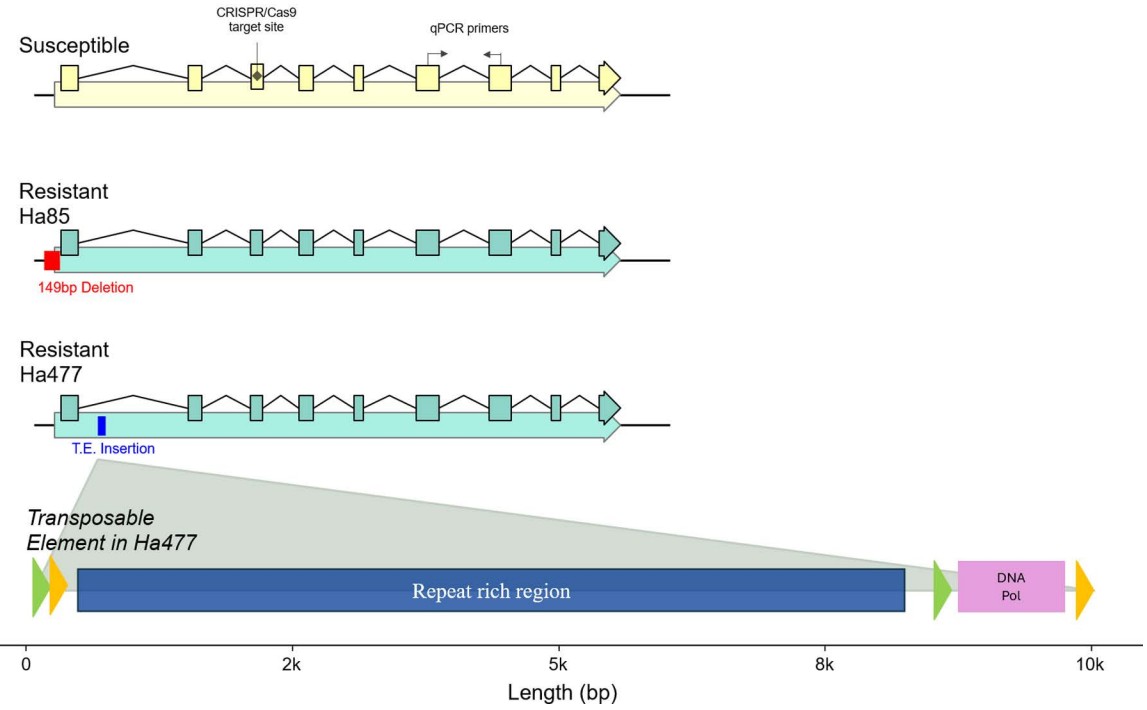

**Fig 4. Genetic variation of the HaVipR1 gene in susceptible and resistant *Helicoverpa armigera* lines.** The CRISPR/Cas9 cut sites along with the locations of qPCR primers are marked on the gene model of the susceptible line. A 149 bp deletion in the 5′ untranslated region (UTR) of the resistant Ha85 line is highlighted with a red box, indicating the genetic alteration contributing to resistance. The insertion site of a transposable element in the resistant Ha477 line is marked with a blue box, showing the site of genomic integration. Detailed representation of the large transposable element features two distinct regions: a large repeat-rich area (coloured in dark blue) and a region with high sequence similarity to a DNA polymerase. Repeat elements and orientation are indicated with orange and green arrows. The transposable element and gene models are drawn to scale.

organisms did not lead to a difference in Vip3Aa resistance [30]. Studies of Vip3Aa toxicity in Sf9 cell lines point toward an increase in autophagy, which appears to increase resistance [17], as well as changes in apoptosis relating to lysosomes and mitochondria function [31]. Indeed, some of these pathways appear to be activated by other compounds (such as the chemical pesticide azadirachtin [32] or harmine [33]) and so be indicative of a general cellular response to internalized toxins, rather than specific to the mechanism of action of Vip3Aa. While our results indicate HaVipR1's involvement in resistance, the specific molecular pathway and its potential interactions with other cellular responses to Vip3Aa remain to be fully elucidated. These findings suggest that HaVipR1 may represent an additional pathway involved in Vip3Aa resistance in the larval midgut, though further research is needed to characterize the mechanisms with relation to other known resistance mechanisms.

The function of HaVipR1, and the mechanism by which its absence leads to Vip3A resistance, is difficult to discern. The gene is widespread across insects and in Lepidoptera generally contains two thyroglobulin domains and a calcium-binding domain (see Fig 3). The disruption of this gene in the resistant line appears to have some fitness cost in the absence of selection (see Fig G in S1 Text, S6 Data). The thyroglobulin domain has been understood to function as a potent cysteine protease inhibitor, specifically of Cathepsins B and L [34,35]. One potential resistance mechanism may be that disruption of this inhibitor leads to an increase in protease activity in the midgut. Activation of Vip3A in the mid-gut requires processing from the inactivated proto-toxin form to the activated toxin by mid-gut proteases [26]. The increased activity of proteases may lead to more rapid degradation of activated Vip3A, enabling resistance. However, previous work characterizing the resistant strain Ha85 identified the opposite case, where the mid-gut juices from the resistant line had a decreased

proteolytic processing of the proto-toxin to the activated toxin, and after 60 min both the resistant and susceptible strain had identical levels of activated Vip3A toxin [36]. In addition, proteases that activate Vip3A are generally serine proteases, not cysteine proteases, and generally serine proteases have been identified to be differentially expressed in response to Vip3A treatment [37]. The identified domains of *HaVipR1* limit its potential role in directly mediating proteolytic activity of proteases on Vip3A, however, further investigations to conclusively exclude this mechanism are required.

Could there be a role for larval resistance derived from changes to a possible receptor that mediates Vip3A binding, beyond that possibly suggested by the Sf9 results discussed above? Changes in receptor binding are highly relevant in mediating resistance in Cry Bt toxins [38], and there is in fact now one report of a change in binding associated with Vip3A resistance in *Helicoverpa zea* [39]. Previously, however, analysis of Brush Border Membrane Vesicles (BBMVs) derived from *H. armigera* resistant and susceptible lines failed to identify any statistically significant differences in the binding of Vip3A [36]. Assessment of the generated knockout line of *HaVipR1* for changes in binding was not carried out in this study, and further investigation of knockout lines of this gene may provide insights into how disruption leads to resistance. While changes were not seen in binding to BBMVs derived from the resistant line, a reduction in the peritrophic matrix (PM) were observed in the case of the chitin-synthase Vip3A resistance in *S. frugiperda*, indicating a potential interaction of Vip3A with other areas of the mid-gut epithelium, or other mid-gut cell types [23]. It is unknown whether the presence of *HaVipR1* in the extracellular space allows a potential interaction with the PM; while changes in permeabilization of the periotrophic matrix have been indicated to be associated with pathogen resistance in *H. armigera*, this was facilitated by a protein able to interact with the PM [40].

The presence of both thyroglobulin and calcium binding domains of HaVipR1 suggests a possible mechanism for resistance, potentially involving pore repair formed by activated Vip3A (outlined in Fig H in S1 Text). Cysteine proteases, specifically cathepsin B and L are involved in targeting repair of pores in cell culture in human cell lines [41]. In addition, calcium ions are a key signal for initiating plasma membrane repair [42]. In its native role, *HaVipR1* may act to regulate the degree of pore repair occurring for damaged cells, helping balance the turnover of mid-gut cells between apoptosis and autophagy [43]. Disruption of *HaVipR1* gene expression may lead to increased repair of pores generated by active Vip3A, and potentially shifts the balance of mid-gut cells from apoptosis to autophagy [44]. In this context, it is noteworthy that the Myb transcription factor, knockout of whose expression in *S. frugiperda* confers resistance to Vip3A [22], drives expression of Bcl2 [45], an apoptosis inhibitor which also affects autophagy [46]. While this potential mechanism aligns with the currently described domains of HaVipR1, additional research is needed to explain the apparent specificity of the resistance and lack of cross-resistance observed, as increased pore repair would be a general mechanism of Bt toxin resistance. The specificity of this resistance mechanism may be explained by differences in ion selectivity between Cry and Vip pores [47]. However, further work exploring the cellular location of this gene, and any potential interaction partners, will be required before this potential mechanism of action can be validated, although some indications exist that repair mechanisms may be involved in Bt resistance [48]. Further research into this area, including cross-resistance studies and binding assays with resistant lines, will be important for understanding the mechanism of action of Vip3A and HaVipR1's role in resistance.

The two identified resistant lines, Ha85 and Ha477, exhibit distinct genetic alterations leading to the disruption of the *HaVipR1* gene and are resistant to Vip3Aa at the highest tested level of Vip3Aa (220 μg/cm$^2$). In Ha85, a small deletion immediately upstream of the start codon appears sufficient to disrupt gene function presumably through disruption of the 5′ region and Kozak sequence, while Ha477 harbors a large transposable element insertion within the first intron sequence, which disrupts regular splicing of this gene (see Fig B in S1 Text). Transposable elements have been increasingly implicated in resistance to various Bt toxins, including Cry1Ac [49], Cry2Ab [50,51], and more recently, Vip3A [23]. The transposable element identified in Ha477 is the largest reported so far. Its classification remains unclear due to the absence of recognized open reading frames that match previously identified elements. It is potentially a composite transposable element, comprising a Class II DNA transposon with an additional insertion of a Class I Retrotransposon

[52]. The large size and complex structure of this transposable element pose significant challenges for identification using short-read sequencing technologies. Repetitive regions complicate effective short-read mapping, emphasizing the need for long-read sequencing to accurately characterize such elements. This has important implications for molecular field monitoring applications, where long-read sequencing would be crucial for identifying resistance alleles in natural populations. These transposable element insertions highlight the complexity of the mutations which contribute to resistance and underscore the necessity of utilizing advanced genomic tools in resistance monitoring and management. The information that transposable elements in introns of genes can dysregulate function provides further evidence of the importance of the identification of transposable elements in Vip3A resistance, and will aid in developing more effective monitoring strategies for combating resistance and ensuring the sustainability of Bt crops.

The level of resistance to Vip3Aa in Australian *H. armigera* populations has remained consistently low, even with the introduction of Bolgard III (which includes Vip3Aa) nearly a decade ago [8]. This is somewhat surprising given the near-complete Vip3Aa resistance observed in field-selected and knockout strains in the present study. While mutations in *HaVipR1* confer strong resistance, the disruption of this gene does appears to incur a fitness cost in the absence of selective pressure (see Fig G in S1 Text, S6 Data). In Australia, robust integrated resistance management (IRM) practices - including the uniform implementation of Bolgard III, mandated non-Bt refuges, and stringent rotation protocols - apply significant selective pressure on *H. armigera* [1]. This combination of strategies leads to "redundant killing," effectively eliminating any *H. armigera* individuals that evolve resistance to a single Bt toxin [53,54]. $F_2$-based monitoring of resistance alleles from the field continues to show detectable Vip3Aa resistant individuals allelic to the Ha85-/Ha477-resistant lines, but with overall frequencies that remain low. The continued low frequency of resistance alleles suggests the current management approach may be contributing to resistance containment, though additional research would be valuable to fully understand the factors maintaining low resistance frequencies. Previous modeling evaluating the effectivness of Vip3Aa under a number of scenarios supports the above conclusions, with the main indication for failure being an increase in resistance to Cry toxins, leading to an change in selection pressures under which the 'redudant killing' approach may not be sufficient [55]. Australian cotton serves as a compelling case study for the effective management of resistance through comprehensive and proactive strategies [1]. This approach can offer valuable insights into sustainable Bt crop management worldwide.

In summary, our study identifies *HaVipR1* as gene associated with Vip3Aa resistance in *H. armigera* and characterizes two instances of disruptive mutation present for this gene. The involvement of transposable elements further complicates the resistance landscape and underscores the necessity for advanced sequencing technologies in resistance monitoring and management. This work contributes to our understanding of potential resistance mechanisms in field-derived lines, which may inform future resistance monitoring approaches. Further research combining empirical data and modeling will be valuable for developing comprehensive resistance management strategies.

## Materials and methods

### Vip3A protein expression

The Vip3A protein utilized in this study originated from three sources. The variant for fitness, monitoring, and bioassay tasks was a Vip3Aa clone provided under a Material Transfer Agreement by Bayer/BASF. For resistance allele identification and subsequent characterization, Vip3Aa sequences were isolated from *Bacillus thuringiensis* isolates maintained at the CSIRO Black Mountain Science Innovation Precinct. The genomes of these isolates were sequenced using an Illumina MiSeq system and assembled via CLC Genomics software. Vip3Aa sequences were identified using BLAST, with specific primers designed to amplify these sequences. Amplified PCR products were cloned into the pGEX4T-1 vector and expressed in *Escherichia coli*. The Vip3Aa used in the validation of the CRISPR-knockout was provided by the Institute of Plant Protection, Chinese Academy of Agricultural Sciences (CAAS), Beijing, China.

## Insect lines, rearing, and bioassay

Insect lines used for linkage mapping, fitness testing, and resistance monitoring, specifically Ha85 and Ha477, are Vip3A-resistant *H. armigera* lines derived from field populations. These lines have been detailed in previous publications [36] and have been maintained and reselected in a laboratory in Canberra, Australia, since 2010. The Ha85 line was backcrossed into the susceptible GR line at least five times, achieving an approximate 92% allelic similarity with the susceptible line. Both the Ha85 and Ha477 lines had high Vip3Aa resistance, characterized by survival of more than 90% of individuals on the maximum amount of toxin able to be applied for a surface treatment (220 µg/cm$^2$). $LC_{50}$ of the susceptible line was $0.551 \pm 0.37$–$0.765$ µg/cm$^2$. Bioassays in this study to test resistance were conducted using a discriminating dose of 10 µg/cm$^2$ of Vip3A, applying the toxin to the surface of the diet (approach based on [56]). Outcomes were evaluated based on mortality after 7 days.

To assess the fitness cost associated with Vip3A resistance, a population cage experiment was conducted, following a protocol similar to [57]. The initial experimental setup included 60 individuals from each population: the Vip3A-resistant population (Ha85) and the susceptible population (GR). These populations were mixed to achieve an initial allele frequency of 50% and were maintained in two replicates for eight generations.

Both populations were not subjected to Vip3A selection pressure throughout the experiment. Over eight generations, a subset of individuals from each replicate population was phenotypically screened using the same Vip3A discriminating dose as described above. The proportion of Vip3A resistance is shown for each generation in Fig G in S1 Text.

## Linkage mapping of resistance allele

**Female informative cross.** Using the absence of recombination during meiosis in female Lepidoptera, the Vip3A resistant locus was mapped to a specific linkage group. This was achieved using a cross involving an Ha85 family (family F2031; Ha85 Vip3A R+/R+ ♂ x GR Vip3A R-/R- ♀). Individual $F_1$ females were subsequently backcrossed to resistant Ha85 males, with offspring screened for resistance using a discriminating dose of Vip3A (10 µg/cm$^2$). For control, over 100 $F_2$ offspring were not bio-assayed, but were included in subsequent genotyping efforts.

**DNA extraction and RAD-seq analysis.** Parents from both crosses and selected and control offspring were preserved in 100% ethanol at −20°C. DNA extraction was performed using the Qiagen Blood and Tissue Kit (Qiagen, the Netherlands) following the manufacturer's instructions. Restriction site Associated DNA Sequencing (RAD-seq) libraries were generated for the female informative cross, following the protocol outlined by [58], utilizing PstI as the restriction enzyme. RADtag libraries were sequenced on an Illumina HiSeq platform at the Biological Resources Facility (BRF), Australian National University, Canberra, Australia. Resulting sequences were processed using STACKS software (v2.59) [59], aligned to the chromosomal genome (Ha2SCD, GCF_023701775.1), and population statistics plotted for valid loci in the selected and control populations for the female informative cross.

## RNA sequencing and transcriptome analysis of resistant lines

RNA was isolated from the midguts of *H. armigera* using Trizol reagent (Life Sciences, Carlsbad, CA, USA) according to the manufacturer's instructions. For transcriptomic analysis, three replicates of pooled midguts from the Vip3Aa-resistant lines (Ha85 and Ha477) and the susceptible line (GR) were prepared, each replicate consisting of RNA extracted from five pooled midguts. RNA sequencing was performed by BGI (Shenzhen, China) using Illumina paired-end technology. Raw sequence data are available under BioProject PRJNA1119665.

The reads were mapped to the *H. armigera* chromosomal genome (HaSCD2, GCF_023701775.1) using Hisat2 (v2.2.1; [60]). Genome-guided transcriptome assemblies were built for each sample with StringTie [61], and a single transcriptome table was generated using StringTie merge, summarized to counts per gene per sample. Proportion of reads mapping to reference and total read counts are presented in Table B in S1 Text.

The resulting gene count table was processed using DESeq2 (v1.44.0; [62]) for differential expression analysis between the Ha85 and GR strains. DESeq2's negative binomial model was used to normalize for library size and compare gene expression across conditions. The default test in DESeq2 was used to generate Wald statistics, and fold change estimates were refined with log fold change shrinkage using the apeglm method [63].

### qPCR confirmation of down-regulated *HaVipR1* gene

Primers were designed to target the differentially regulated candidate gene identified in the RNAseq analysis. Association of the resistance allele with the candidate transcript was evaluated through PCR in various samples, including individuals from the Ha85 mapping family, the Ha477 resistant colony, a double-resistant line ("DRES") derived from Ha477 and SP15 (a Cry2Ab-resistant line), and families identified as resistant in the Vip3A resistance monitoring project. For the qPCR analysis, midguts from late 3rd instar larvae of *H. armigera* lines GR, Ha85, and Ha477 were dissected and snap-frozen in liquid nitrogen. RNA extraction from the midguts was performed using the RNeasy Mini Kit (QIAGEN, Hilden, Germany, Catalog No. 74104). We employed the Luna Universal One-Step RT-qPCR Kit for the qPCR reactions (New England Bio-labs, Ipswich, MA, USA, Catalog No. E3005). Each 20 µL reaction consisted of 10 µL reaction mix, 1 µL enzyme mix, 1.6 µL each of forward and reverse primers, 2 µL RNA (diluted to 150 ng/µL), and 5.4 µL water. The reference gene EF-α was used, with five biological replicates and three technical replicates for each gene. Primer sequences were verified by PCR and Sanger sequencing using 'Big Dye' chemistry. The primer sequences used were as follows:Vip3A primers: VipF2: 5′-GGA AGG GTA CAA CGT GGA CT-3′; VipR2: 5′-CAC GCT CGT CGA TAC AGA TT-3′. Reference gene primers: EF1α Forward: 5′-GAC AAA CGT ACC ATC GAG AAG-3′; EF1α Reverse: 5′-GAT ACC AGC CTC GAA CTC AC-3′. All qPCR reactions and data analysis were conducted on a BioRad real-time PCR system instrument.

### Long-read sequencing and de novo assembly of the resistant lines

High-molecular-weight (HMW) DNA was extracted from the head and thorax (approximately 50 mg of tissue) of an adult moth from the Ha477 and Ha85 lines using the Circulomics HMW Nanobind kit (Circulomics, Catalog No. 102-301-900). DNA extraction quality and quantity were assessed using the Promega QuantiFluor dsDNA system. The extracted DNA was then submitted to the Australian Genome Research Facility (AGRF) for long-read sequencing using Oxford Nanopore Technologies (ONT). The HMW DNA sample was processed for sequencing by AGRF following the PromethION P2 Kit V13 (SQK-LSK113) barcoding protocol and sequenced on a PromethION flow cell (type R10.4.1, FLO-PRO114M). Base-calling was performed directly on the flow cell using High-Accuracy mode. Additionally, a matched Illumina short-read library was generated on a NovaSeq S4. The raw data have been deposited in the SRA under accession number PRJNA1119665.

The long-read data was de novo assembled using Flye (v2.9.3) [64] and polished using the matched short-read data for three rounds with Pilon (v1.24) [65], following the recommended pipeline. The resulting assemblies were then mapped to the reference using minimap2 (v2.25) [66], sorted and indexed using samtools (v1.18) [67]. The region containing the *HaVipR1* gene was extracted from the polished assembly, and the repeat content was analyzed using the Geneious Prime (2023.2.1) Repeat Finder tool.

### Impact of transposable element present in Ha477 line on expression of the HaVipR1 gene

The expression of the HaVipR1 gene appears to be altered in the Ha477 line due to the presence of the large insertion in intron 1, however due to the insertion's large size it is not able to be mapped to the HaSCD2 reference. To assess whether the large insertion was impacting gene expression in Ha477, the RNA-seq data from the susceptible and Ha477 lines were aligned to the polished Ha477 assembly using Hisat2 and splicing was assessed visually using the Integrated Genomics Viewer (IGV, v 2.17) and is shown in Fig B in S1 Text.

## Phylogenetic analysis of the *HaVipR1* gene

The *HaVipR1* gene (LOC110373801/ XP_063898125.1) was employed as a query sequence for a BlastP search against the NCBI non-redundant (nr) protein sequences database. The top 100 hits were retrieved, and the resulting protein sequences were curated to remove duplicates and to retain only a single representative per species in cases of multiple hits from different genome assemblies, resulting in the exclusion of 13 genes.

The selected protein sequences were aligned using the online MUSCLE tool (version 3.8.425), employing default settings. This alignment was then used to construct a phylogenetic tree using the IQ-TREE2 web server [68,69]. The tree was built with 1,000 ultrafast bootstrap replications, and the best-fit model (WAG+R5) was selected based on the Bayesian Information Criterion. One sequence (XP_013177522.1/PREDICTED: uncharacterized protein LOC106124990 from *Papilio xuthus*) was excluded from the analysis due to its having over 50% gaps compared to other sequences. Ultimately, 86 of the initial 100 protein sequences were included in the final tree analysis.

Visualization of the phylogenetic tree was conducted using the ggtree package [70], with bootstrap support values annotated on each node. The multiple sequence alignment file, used as input, is provided with amino acid residues coloured by type. Taxonomic classifications for each species were extracted from the NCBI Taxonomy database using a custom Python script.

## CRISPR/Cas9 knockout of the *HaVipR1* gene

The wild-type strain SCD that was originally collected from Côte D'Ivoire (Ivory Coast, Africa) in the 1970s was kindly provided by Bayer Crop Science in 2001 [71]. This strain has been maintained in the laboratory without exposure to insecticides or Bt toxins for over 30 years, and it is susceptible to both Bt toxins and chemical insecticides [71]. The *HaVipR1* gene (LOC110373801; HaOG214548) of the SCD strain was knocked out with the CRISPR/Cas9 genome editing tool [27] to produce a knockout strain homozygous for a 16-bp deletion in exon 3 of HaVip3R1.

## Vip3Aa Toxicity and Bioassays of the CRISPR/Cas9 knockout strain

The Vip3Aa protoxin was provided by the Institute of Plant Protection, Chinese Academy of Agricultural Sciences (CAAS), Beijing, China. Protoxin in solution was prepared by diluting the stock suspensions with a 0.01 M, pH 7.4, phosphate buffer solution. Liquid artificial diet (1,000 µl) was dispensed into each well (surface area = 2 cm$^2$) of a 24-well plate. After the diet cooled and solidified, 100 µl of the Vip3Aa protein solution was applied evenly to the diet surface in each well. A single unfed neonate (24-h old) was put in each well after the Bt protein solution had dried at room temperature. Mortality was recorded after 7 days. Larvae were considered as dead if they died or weighed less than 5 mg at the end of bioassays.

The sgRNA target sequences (5′-GGCACCACGATAGGAAGATGTGG-3′, underlined is the PAM sequence) were selected at exon 3 of *HaVipR1* according to the principle of 5′-GGN18NGG-3′ (Fig A in S1 Text). The template DNA was made with PCR-based fusion of two oligonucleotides: one is the specific oligonucleotide (CRISPR-F) encoding T7 polymerase-binding site and the sgRNA target sequence GGN18, and the other is the universal oligonucleotide (sgRNA-R) encoding the remaining sgRNA sequences. The fusion PCR reaction mixture (50 µl) consisted of 25 µl of PrimeSTAR HS (Premix) (TaKaRa, Dalian, China), 2 µl of 10 µM CRISPR-F, 2 µl of 10 µM sgRNA-R, and 21 µl of ddH2O. PCR was performed at 98°C for 30 s, 35 cycles of (98°C for 10 s, 60°C for 20 s, 72°C for 30 s), followed by 72°C for 10 min, and held at 12°C. The PCR products were then purified (using the QIAquick PCR Purification Kit (QIAGEN, Hilden, Germany). The sgRNA was synthesized by in vitro transcription utilizing the GeneArt Precision gRNA Synthesis Kit (Thermo Fisher Scientific, Shanghai, China) according to the manufacturer's instruction. The Cas9 protein (GeneArt Platinum Cas9 Nuclease) was purchased from Thermo Fisher Scientific (Shanghai, China).

The collection and preparation of eggs were carried out as previously reported [17]. Briefly, fresh eggs laid within 2 h were washed down from the gauzes using a 1% sodium hypochlorite solution and rinsed with distilled water. After

suction filtration, the eggs were lined up on a microscope slide fixed with double-sided adhesive tape. Approximately one nano-liter mix of sgRNA (300 ng/µl) and Cas9 protein (100 ng/µl) were injected into individual eggs using a FemtoJet and InjectMan NI 2 microinjection system (Eppendorf, Hamburg, Germany). The microinjection process was completed within 2 h. Injected eggs were then placed at 26 ± 1°C, 60 ± 10% RH for hatching.

To identify the indel mutations at exon 3 of *HaVipR1*, a pair of specific primers (forward: 5′-AACGTGAGGTTACTCATGCCTT-3′, reverse: 5′-GCCAAGCTTATTTTGACCAGCC-3′) was designed to amplify a ~ 500-bp fragment flanking exon 3. The PCR fragments were amplified from genomic DNA samples of individual insects, and then sent to Life Technology (Shanghai, China) for direct sequencing, using the forward primer used as the sequencing primer. To determine the association between the Vip3Aa resistance phenotype and the 16-bp deletion mutation of *HaVipR1*, we performed a chi-squared test of independence. Observed genotype frequencies for both treated (survivors) and untreated groups (control) were compared to expected frequencies under the assumption of genotype and treatment independence. Expected frequencies were calculated based on the total counts of each genotype across both groups.

The resistance level of the susceptible SCD strain and the generated knockout strain (548KO2) was assessed with a surface treatment bio-assay as described in [8]. The total number of larvae tested for both the SCD and the knockout strain was 480.

To determine the inheritance pattern of Vip3Aa resistance, reciprocal crosses were established between the resistant 548KO2 strain and susceptible SCD strain. Thirty virgin moths from each strain were used for reciprocal crosses (resistant female × susceptible male, and susceptible female × resistant male). Neonates from the parental strains (548KO2 and SCD) and their F1 progeny were tested using surface-treated artificial diet bioassays. For each strain and cross, 96 neonates were exposed to a diagnostic concentration of 4 µg Vip3Aa/cm². Mortality was assessed after 7 days. The degree of dominance ($h$) was calculated according to [72] using the formula: $h = (W_{12} - W_{22})/(W_{11} - W_{22})$, where $W_{11}$, $W_{12}$, and $W_{2}2$ are the fitness values of resistant homozygotes, heterozygotes, and susceptible homozygotes, respectively. Complete survival was scored as 1 and complete mortality as 0.

## Supporting information

**S1 Text.  Fig A: Schematic diagram and genotyping of the HaVip3R1 gene edited by CRISPR/Cas9.** Description: (A) Schematic representation of the sgRNA-targeted site and sequences in exon 3 of HaVip3R1. (B) Indel mutations in the G1 larvae from single-pair families between G0 and SCD, highlighting the target sequences of the wild-type HaVip3R1 allele and the CRISPR/Cas9-induced mutations. (C) Representative chromatograms showing the 16-bp deletion in HaVip3R1 after CRISPR/Cas9 editing. **Fig B: Sashimi plot showing splicing disruption in the HaVipR1 transcript in the Ha477-resistant line.** Description: Sashimi plot illustrating the disruption of normal splicing in the HaVipR1 gene in the Ha477-resistant line, caused by a transposable element in intron 1. Data were obtained from pooled RNA-seq from susceptible and resistant *H. armigera* lines, showing abnormal splicing in the resistant line. **Fig C: Expression analysis of 8 Vip3A-related genes from *Spodoptera frugiperda* in H. armigera mid-gut transcriptome.** Description: Variance-stabilized expression values for eight Vip3A-related genes in susceptible and resistant allelic (Ha85 and Ha477) lines of *H. armigera*. Statistical significance was calculated using a Student *t* test, and p-values are provided for each comparison. Data available in S2 Data. **Fig D: Spatio-transcriptome expression of HaVipR1 in *H. armigera*.** Description: Expression values of HaVipR1 across different stages (L2, L3, and L4) and gut compartments (L5) in *H. armigera*. The data were obtained from the spatio-transcriptome dataset of *H. armigera*, including standard error for each condition. **Fig E: Expression of the HaVipR1 homologue in *Bombyx mori* midgut.** Description: Expression analysis of the HaVipR1 homologue in *Bombyx mori* midgut, using data from the SilkBase database. The homologous gene, LOC101735440, was found to have peak expression in the midgut. Data available in S4 Data. **Fig F: Expression of HaVipR1 in *Spodoptera frugiperda* mid-gut and Sf9 cell lines.** Description: Analysis

of HaVipR1 expression in *Spodoptera frugiperda* mid-gut tissues and Sf9 cell lines. Data were obtained from aligned RNA-seq samples as described in Table S4. Data available in S5 Data. **Fig G: Decrease in HaVipR1 phenotype over time in the absence of selection.** Description: Decline in the frequency of the Vip3A resistance phenotype in the Ha85-resistant line over 8 generations in the absence of selection. The data were collected from two replicates (X1 and X2). Data available in S6 Data. **Fig H: Proposed mechanism of Vip3Aa resistance through disruption of HaVipR1.** Description: Schematic diagram depicting the proposed mechanism of Vip3Aa resistance through the disruption of HaVipR1, hypothesized to play a role in midgut membrane repair. **Table A: Number of RadTag loci per chromosome in female informative cross.** Description: List of RadTag loci identified across chromosomes in the female informative cross. Each chromosome's identifier and the number of loci are provided. **Table B: Mapping statistics for transcriptome samples from resistant (Ha85 and Ha477) and susceptible (GR) *H. armigera* lines.** Description: Mapping statistics for RNA-seq samples from three strains of *H. armigera*: Ha85, Ha477, and GR (susceptible). The total reads and the percentage of reads mapped to the reference genome are listed for each sample. **Table C: Read count and DeSeq2 statistics for HaVipR1 gene analysis between the Ha85-resistant line and the GR susceptible line.** Description: Raw read counts and statistical analysis from DeSeq2 for HaVipR1 expression in the Ha85-resistant line and the GR susceptible line. Includes p-values for the comparison. **Table D: Reciprocal best hit analysis of *S. frugiperda* and *H. armigera* genes.** Description: Results from reciprocal best-hit analysis of genes related to Vip3A between *Spodoptera frugiperda* (Sf) and *Helicoverpa armigera* (Ha). Includes gene names, identifiers, and short names. **Table E: SRA identifiers for Sf9 and *Spodoptera frugiperda* mid-gut samples analyzed for HaVipR1 expression.** Description: A list of SRA (Sequence Read Archive) identifiers for samples from Sf9 cell lines and *S. frugiperda* mid-gut tissues used to analyze HaVipR1 expression. Each entry corresponds to a specific sample. (DOCX)

**S1 Data. Fst and Φst values from STACKS analysis of Vip3A-resistant and susceptible backcross progeny.** Values for Fst and Φst calculated in 1 Mbp genomic windows across all chromosomes from STACKS analysis. Data were derived from RAD-Seq analysis of progeny from a female-informative backcross, comparing individuals selected on Vip3A versus unexposed controls. (XLSX)

**S2 Data. RNA-Seq and qPCR data comparing gene expression in Vip3A-resistant and susceptible lines. Data file of raw RNA-Seq read counts for all genes across three strains (Ha85, Ha477, GR), differential expression results, and qPCR Cq values.** (XLSX)

**S3 Data. Summary of transcriptomic and proteomic data across developmental stages for *Helicoverpa armigera*. Originally generated by [73].** (XLSX)

**S4 Data. Expression values from the SilkWorm database for the *Bombyx mori* homologue of HaVipR1.** (XLSX)

**S5 Data. Expression counts for the *Spodoptera frugiperda* HaVipR1 homologue (LOC118272819) in mid-gut and Sf9 cell lines.** (XLSX)

**S6 Data. Results from population cage experiment of Vip3Aa resistant colonies in the absence of selection.** (XLSX)

## Author contributions

**Conceptualization:** Craig J. Anderson, Yiyun Wei, Yidong Wu, Sharon Downes, Ashley E. Tessnow, Gregory A. Sword, Michelle Williams, Wee Tek Tay, Karl H. J. Gordon, Tom K. Walsh.

**Data curation:** Andreas Bachler, Amanda Padovan, Craig J. Anderson, Yiyun Wei, Yidong Wu, Sharon Downes, Bill James, Gregory A. Sword, Karl H. J. Gordon, Tom K. Walsh.

**Formal analysis:** Andreas Bachler, Amanda Padovan, Craig J. Anderson, Yiyun Wei, Yidong Wu, Stephen Pearce, Sharon Downes, Ashley E. Tessnow, Gregory A. Sword, Michelle Williams, Wee Tek Tay, Karl H. J. Gordon, Tom K. Walsh.

**Funding acquisition:** Amanda Padovan, Yidong Wu, Gregory A. Sword, Karl H. J. Gordon, Tom K. Walsh.

**Investigation:** Andreas Bachler, Amanda Padovan, Craig J. Anderson, Yiyun Wei, Yidong Wu, Stephen Pearce, Sharon Downes, Ashley E. Tessnow, Michelle Williams, Karl H. J. Gordon, Tom K. Walsh.

**Methodology:** Andreas Bachler, Amanda Padovan, Craig J. Anderson, Yiyun Wei, Yidong Wu, Stephen Pearce, Sharon Downes, Bill James, Ashley E. Tessnow, Gregory A. Sword, Michelle Williams, Wee Tek Tay, Karl H. J. Gordon, Tom K. Walsh.

**Project administration:** Amanda Padovan, Yidong Wu, Sharon Downes, Gregory A. Sword, Wee Tek Tay, Karl H. J. Gordon, Tom K. Walsh.

**Resources:** Amanda Padovan, Yidong Wu, Sharon Downes, Bill James, Michelle Williams, Karl H. J. Gordon, Tom K. Walsh.

**Software:** Karl H. J. Gordon.

**Supervision:** Amanda Padovan, Yidong Wu, Bill James, Gregory A. Sword, Wee Tek Tay, Karl H. J. Gordon, Tom K. Walsh.

**Validation:** Andreas Bachler, Craig J. Anderson, Yiyun Wei, Stephen Pearce, Sharon Downes, Ashley E. Tessnow, Michelle Williams, Tom K. Walsh.

**Visualization:** Andreas Bachler.

**Writing – original draft:** Andreas Bachler, Amanda Padovan, Tom K. Walsh.

**Writing – review & editing:** Andreas Bachler, Amanda Padovan, Craig J. Anderson, Yiyun Wei, Yidong Wu, Stephen Pearce, Sharon Downes, Ashley E. Tessnow, Gregory A. Sword, Wee Tek Tay, Karl H. J. Gordon, Tom K. Walsh.

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
