## [Editor Report · Decision Letter 0]

28 Jan 2025

Dear Dr Bachler, 

Thank you for submitting your manuscript entitled "Disruptions of a novel gene confer Vip3Aa resistance in two field-derived lines of Helicoverpa armigera" for consideration as a Research Article by PLOS Biology.

Your manuscript has now been evaluated by the PLOS Biology editorial staff, as well as by an academic editor with relevant expertise, and I am writing to let you know that we would like to send your submission out for external peer review.

IMPORTANT: After discussions with the editorial team, we would like to consider your manuscript as a 'Short Report' at the journal. Upon resubmission (see details below), I would be grateful if you could please tick 'Short Report' as the article type in the dropdown menu. In addition, Short Reports have a maximum of 4 main figures, so please reduce the number of main figures by 1 (either by moving to the Supplementary or by combining two figures into 1).

Before we can send your manuscript to reviewers, we need you to complete your submission by providing the metadata that is required for full assessment. To this end, please login to Editorial Manager where you will find the paper in the 'Submissions Needing Revisions' folder on your homepage. Please click 'Revise Submission' from the Action Links and complete all additional questions in the submission questionnaire.

Once your full submission is complete, your paper will undergo a series of checks in preparation for peer review. After your manuscript has passed the checks it will be sent out for review. To provide the metadata for your submission, please Login to Editorial Manager (https://www.editorialmanager.com/pbiology) within two working days, i.e. by Jan 30 2025 11:59PM.

Kind regards,

Richard

Richard Hodge, PhD

rhodge@plos.org

PLOS

---

## [Editor Report · Decision Letter 1]

14 Feb 2025

Dear Dr Bachler,

Thank you for your continued patience while we considered your revised manuscript "Disruptions of a novel gene confer Vip3Aa resistance in two field-derived lines of Helicoverpa armigera" for publication as a Short Report at PLOS Biology. Your revised study has been evaluated by the PLOS Biology editors and the Academic Editor.

Our Academic Editor has now arbitrated the previous reviews received at PNAS and the rebuttal to avoid another round of peer review and loss of time. I have pasted some specific feedback from the Academic Editor below my signature (labelled ‘Comments from the Academic Editor’). As you will see, the Academic Editor appreciates that many of the previous comments have been addressed and clarified. However, he/she notes that additional data to characterize the heterozygous lines (from crosses between Vip3A-resistant and susceptible lines) has not been provided and we ask that dose-response data is included in a revised version. On the other hand, we think that new data to explain why HaVipR1 resistance alleles have not been favoured by selection, or identifying a definitive mechanism of action, is not required in the Short Report format. Instead, we ask that some overstatements (as noted in the AE comments) are toned down in the manuscript text. 

Given the extent of revision needed, we cannot make a decision about publication until we have seen the revised manuscript and your response to the reviewers' comments. Your revised manuscript is likely to be sent for further evaluation by all or a subset of the reviewers.

**IMPORTANT - SUBMITTING YOUR REVISION**

*Re-submission Checklist*

*Published Peer Review*

*PLOS Data Policy*

*Blot and Gel Data Policy*

Best regards,

Richard

Richard Hodge, PhD

rhodge@plos.org

COMMENTS FROM THE ACADEMIC EDITOR

The authors have addressed many of the previous editor’s and reviewers’ comments and clarified their methods, however a few important issues remain only partially resolved. They have not generated new dose-response data for heterozygotes to confirm whether resistance is recessive at all toxin concentrations, nor have they conducted cross-resistance or binding assays in the knockout line to confirm that HaVipR1-mediated resistance aligns with field-derived resistant lines. Although the original knockout line is no longer available for cross-resistance or binding experiments, characterizing heterozygotes remains essential.

It remains unclear whether HaVipR1 encodes a Vip3Aa receptor or performs a more general detoxification/metabolic role, and accordingly the authors have shifted their focus away from a definitive mechanism of action and toward identifying the gene and hypothesizing its possible roles. Nonetheless, they should moderate their claim that “HaVipR1-mediated resistance operates independently of known resistance genes, including midgut-specific chitin synthase and the transcription factor SfMyb.” While the evidence supports the possibility that HaVipR1 represents a novel mechanism of Vip3Aa resistance, the specific molecular pathway remains unknown, so this conclusion cannot be definitively established.

While the authors suggest that integrated resistance management explains the low frequency of HaVipR1 mutations in the field, they do not supply new empirical or modeling data that directly test why HaVipR1 resistance alleles have not been favored by selection. The authors acknowledge that modeling and further field data could clarify why resistance alleles remain rare but believe existing theoretical and empirical work already supports the current observations. They emphasize that, while HaVipR1 is an important resistance gene, comprehensive monitoring, including other potential mechanisms, is crucial to managing Vip3Aa resistance. However, it is an overstatement to claim in the abstract that “identifying HaVipR1 forms a foundation for assessing resistance management strategies to preserve Bt crop efficacy.

---

## [Editor Report · Decision Letter 2]

7 Mar 2025

Dear Andy,

Thank you for your patience while we considered your revised manuscript "Disruptions of a novel gene confer Vip3Aa resistance in two field-derived lines of Helicoverpa armigera" for publication as a Short Report at PLOS Biology. 

I have now discussed the revision with our Academic Editor and their comments are provided are below my signature (labeled as ‘Comments from the Academic Editor’). While we appreciate the new data to assess dose-dependent resistance for the heterozygous knockouts, the Academic Editor raises concerns that only a single concentration (4 µg/cm2) was tested and a full dose-response curve has not been generated. While we recognize that follow up experiments are more challenging given that the HaVipR1 knockout line is no longer available, we ask that a broader range of doses are assessed to fully support the claims. 

Given the extent of revision needed, we cannot make a decision about publication until we have seen the revised manuscript and your response to the Academic Editor's comments. 

**IMPORTANT - SUBMITTING YOUR REVISION**

*Re-submission Checklist*

*Published Peer Review*

*PLOS Data Policy*

*Blot and Gel Data Policy*

Best regards,

Richard

Richard Hodge, PhD

rhodge@plos.org

COMMENTS FROM THE ACADEMIC EDITOR 

1. The authors have not generated a full dose-response curve for heterozygous knockouts to confirm whether resistance is recessive at all toxin concentrations. While the newly added data indicate that resistance is recessive at 4 μg/cm², assessing a broader range of doses is necessary for a robust conclusion.

2. The authors could not directly compare the resistance ratios of the knockout line to those of the Ha85 and Ha477 lines, making it unclear whether additional genes beyond HaVipR1 might contribute to resistance in field-derived populations. The LC50 could not be established, as the maximum tested concentration (500 μg/cm² for the knockout and 220 μg/cm² for the resistant lines) resulted in less than 50% mortality. Thus, the claim that “the knockout matches the Vip3Aa resistance phenotype identified in both of the field-derived Vip3Aa resistant lines” remains unsupported by the current data.

---

## [Editor Report · Decision Letter 3]

3 Apr 2025

Dear Andy,

Thank you for your patience while we considered your revised manuscript "Disruptions of a novel gene confer Vip3Aa resistance in two field-derived lines of Helicoverpa armigera" for publication as a Short Report at PLOS Biology. This revised version of your manuscript has been evaluated by the PLOS Biology editors, the Academic Editor.

Based on on our Academic Editor's assessment of your revision, I am pleased to say that we are likely to accept this manuscript for publication, provided you satisfactorily address the following data and other policy-related requests that I have provided below (A-D):

(A) We routinely suggest changes to titles to ensure maximum accessibility for a broad, non-specialist readership. In this case, we would suggest a minor edit to the title, as follows. Please ensure you change both the manuscript file and the online submission system, as they need to match for final acceptance:

“Disruption of HaVipR1 confers Vip3Aa resistance in the moth crop pest Helicoverpa armigera”

(B) You may be aware of the PLOS Data Policy, which requires that all data be made available without restriction: http://journals.plos.org/plosbiology/s/data-availability. For more information, please also see this editorial: http://dx.doi.org/10.1371/journal.pbio.1001797

-Supplementary files (e.g., excel). Please ensure that all data files are uploaded as 'Supporting Information' and are invariably referred to (in the manuscript, figure legends, and the Description field when uploading your files) using the following format verbatim: S1 Data, S2 Data, etc. Multiple panels of a single or even several figures can be included as multiple sheets in one excel file that is saved using exactly the following convention: S1_Data.xlsx (using an underscore).

-Deposition in a publicly available repository. Please also provide the accession code or a reviewer link so that we may view your data before publication. 

Figure 1, 2A-C, S3, S5, S6, S7, S8

(C) Please also ensure that each of the relevant figure legends in your manuscript include information on *WHERE THE UNDERLYING DATA CAN BE FOUND*, and ensure your supplemental data file/s has a legend.

(D) Per journal policy, if you have generated any custom code during the course of this investigation, please make it available without restrictions. Please ensure that the code is sufficiently well documented and reusable, and that your Data Statement in the Editorial Manager submission system accurately describes where your code can be found. 

We expect to receive your revised manuscript within two weeks. 

*Published Peer Review History*

*Press*

Best wishes,

Richard

Richard Hodge, PhD

rhodge@plos.org

PLOS

---

## [Editor Report · Decision Letter 4]

16 Apr 2025

Dear Andy,

On behalf of my colleagues and the Academic Editor, Louis Lambrechts, I am pleased to say that we can accept your manuscript for publication, provided you address any remaining formatting and reporting issues. These will be detailed in an email you should receive within 2-3 business days from our colleagues in the journal operations team; no action is required from you until then. Please note that we will not be able to formally accept your manuscript and schedule it for publication until you have completed any requested changes.

PRESS

Best wishes, 

Richard

Richard Hodge, PhD

rhodge@plos.org

PLOS
